# Spatio-Temporal Heterogeneity of Climate Warming in the Chinese Tianshan Mountainous Region

Xuemei Li [1,2,3,*], Bo Zhang [1,2,3], Rui Ren [4], Lanhai Li [5] and Slobodan P. Simonovic [6]

1   Faculty of Geomatics, Lanzhou Jiaotong University, Lanzhou 730070, China; 18235118550@163.com
2   National-Local Joint Engineering Research Center of Technologies and Applications for National Geographic State Monitoring, Lanzhou 730070, China
3   Gansu Provincial Engineering Laboratory for National Geographic State Monitoring, Lanzhou 730070, China
4   Key Laboratory of Regional Climate-Environment in Temperate East Asia, Institute of Atmospheric Physics, Chinese Academy of Sciences, Beijing 100029, China; renrui@tea.ac.cn
5   State Key Laboratory of Desert and Oasis Ecology, Xinjiang Institute of Ecology and Geography, Chinese Academy of Sciences, Urumqi 830011, China; lilh@ms.xjb.ac.cn
6   Institute for Catastrophic Loss Reduction, The University of Western Ontario, London, ON N6A 5B9, Canada; simonovic@uwo.ca
*   Correspondence: lixuemei@lzjtu.edu.cn; Tel.: +86-17794268162

**Abstract:** The Chinese Tianshan mountainous region (CTMR) is a typical alpine region with high topographic heterogeneity, characterized by a large altitude span, complex topography, and diverse landscapes. A significant increase in air temperature had occurred in the CTMR during the last five decades. However, the detailed, comprehensive, and systematical characteristics of climate warming, such as its temporal and spatial heterogeneity, remain unclear. In this study, the temporal and spatial heterogeneity of climate warming across the CTMR had been comprehensively analyzed based on the 10-day air temperature data gathered during 1961–2020 from 26 meteorological stations. The results revealed local cooling in the context of general warming in the CTMR. The amplitude of variation (AV) varied from −0.57 to 3.64 °C, with the average value of 1.19 °C during the last six decades. The lapse rates of the elevation-dependent warming that existed annually, and in spring, summer, and autumn are −0.5 °C/100 m, −0.5 °C/100 m, −0.7 °C/100 m, and −0.4 °C/100 m, respectively. The warming in the CTMR is characteristic of high temporal heterogeneity, as represented by the amplified warming at 10-d scale for more than half a year, and the values of AV were higher than 1.09 °C of the global warming during 2011–2020 (GWV$_{2011-2020}$). Meanwhile, the amplitudes of warming differed greatly on a seasonal scale, with the rates in spring, autumn, and winter higher than that in summer. The large spatial heterogeneity of climate warming also occurred across the CTMR. The warming pole existed in the warm part, the Turpan-Hami basin (below 1000 m asl) where the air temperature itself was high. That is, the warm places were warmer across the CTMR. The cooling pole was also found in the Kuqa region (about 1000 m asl). This study could greatly improve the understanding of the spatio-temporal dynamics, patterns, and regional heterogeneity of climate warming across the CTMR and even northwest China.

**Keywords:** climate warming; alpine region; temporal and spatial heterogeneity; amplitude; Chinese Tianshan mountainous region

## 1. Introduction

Global surface temperature (GST) in the last 20 years (2001–2020) has increased by 0.99 °C relative to 1850–1900. The last 10 years (2011–2020) are regarded as the warmest 10-year period since 1850. GST in 2011–2020 was 1.09 °C higher than that in 1850–1900 [1,2]. Global warming is enhanced dramatically. However, the rate of long-term change and the amplitude of inter-annual variability differed from global to regional to local scales, between regions, and across climate variables [3]. Climate warming has more severe climate

warming, while other areas have milder climate warming, and some places may even cool down. For example, the amplitude of GST over land was 1.59 °C, larger than that over the ocean (0.88 °C) [1]. Both the largest changes in temperature and the largest amplitude of year-to-year variations were observed in the Arctic. Nevertheless, tropical regions had experienced less warming than most others but also exhibited smaller inter-annual variations in temperature [4]. The emergence of warming is more apparent in northern South America, East Asia, and central Africa than in northern North America or northern Europe [5,6]. Europe–west Asia and northeast Asia were regions with amplified summer warming under the background of global warming [6–8].

Meanwhile, the characters of climate warming were not uniform. Warming was more rapid at higher elevations, a majority of studies suggested. Enhanced rates of warming at higher elevations in the Tibetan Plateau–Himalayan region were found in the 20th century, and this phenomenon was projected to strengthen by the end of the 21st century under a high-emission scenario [1,9,10]. A constant lapse rate of both surface air temperature and dew-point temperature along the slope was observed during typical monsoon months along Khumbu Valley, south of Mt. Everest in the central Himalayas [11]. The annual average temperature (AAT) in tropical alpine areas had been detected with a significant elevation-dependent warming (EDW) phenomenon [12]. Eurasian warming featured evident spatial heterogeneity with a zonal tripole pattern [13].

China is a vulnerable, sensitive, and significant region of global warming. From 1951 to 2020, the AAT in China had increased by 1.82 °C (0.26 °C/10a), which was higher than GST over land during the same period [14]. Changes to mean temperature depends on the elevation in the Yellow River basin [15], while the relative long-term observations of permafrost temperatures in the headwater area of the Yellow River showed no EDW [16–18]. For northwestern China, the increased AAT varied from 0.10 to 0.88 °C/10a during 1961–2010, and the first decade of the 21st century was the warmest decade over the last 1000 years. Furthermore, warming was different from year to year and from season to season, with large temporal heterogeneity, with both seasonal and annual warming being observed in northwestern China [19]. Global warming is likely to reach 1.5 °C between 2030 and 2052 if it continues to increase at the current rate [20]. With the increase in global AAT getting closer to the 1.5 °C target, there is an increasing need for research to understand the warming characteristics of different parts of the world on the global warming target.

Owing to the higher sensitivity of glaciers and snow to climate change than other geomorphic types at the same latitude, mountainous regions are regarded as the outposts of global climate change [21–23]. Characterized by large altitude span, complex topography, diverse landscapes, and exposure, mountainous temperatures often suffer from extreme local variability with high topographic heterogeneity [24–27]. The air temperatures in mountainous regions at different altitudes, slopes, or aspects show distinctly different seasonal and annual trends. Therefore, it is necessary to better understand the spatial and temporal heterogeneity of warming caused by topographic heterogeneity or other reasons in mountainous regions such as the Tianshan mountainous region (TMR). The TMR is the largest independent latitudinal mountain system, the farthest mountain system from the ocean, and the largest mountain system in the arid regions of the world. It is extremely important for assessing climate change and the ecological environment in northwestern China, and the whole country, because of its special geographical location and complex terrain. The TMR is approximately 250–350 km wide and over 2500 km long, spanning from Uzbekistan to Kyrgyzstan, southeastern Kazakhstan, and Xinjiang (China) with an area of $8 \times 10^5$ km$^2$ [28–30]. The TMR belongs to Xinjiang, is about $5.7 \times 10^5$ km$^2$ (about 1700 km long), accounts for 34.5% of the total area of Xinjiang Uygur Autonomous Region [31,32], and is referred to as the Chinese TMR (CTMR) in this study (Figure 1). The average height of the mountain ridge is 4000 m asl, with the highest peak, Tomor, reaching 7443.8 m asl. The CTMR is a typical alpine mountainous area characterized by a continental climate with large seasonal differences. Mainly affected by the westerly circulation and topography, the CTMR has relative abundant precipitation, regarded as the main source of

water resources in Xinjiang. The average accumulated precipitation was 165 mm during 1961–2020 with the average increasing rate of 8.8 mm/10a. The warming rate in the CTMR reached 0.32–0.42 °C/10a during the past 53 years (1961–2013), which was much higher than the national average [33,34].

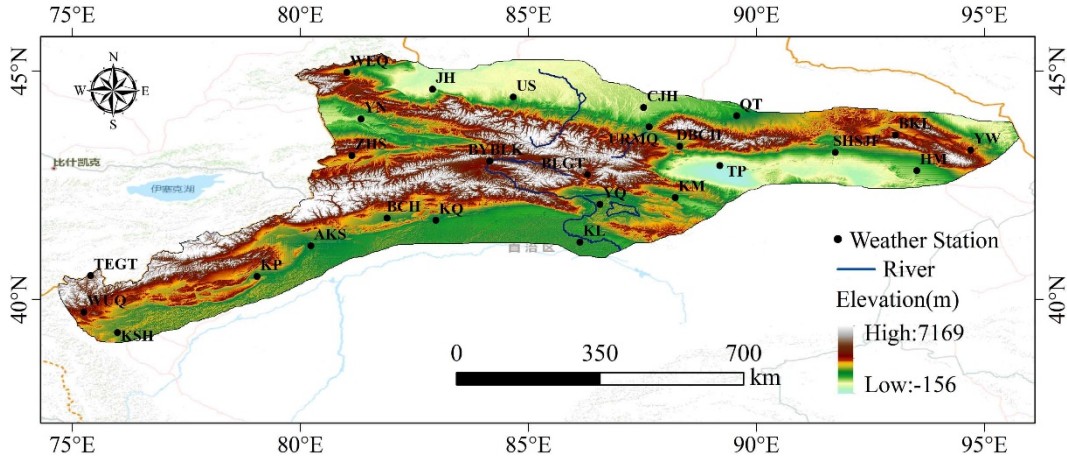

**Figure 1.** Map of the study area.

As the sensitivity of the human society and ecosystem to climate warming varies for different regions, it is meaningful to quantify the amplitude and extent of climate warming in the CTMR. However, current research on climate warming in the CTMR was not detailed and did not show much about the spatial and temporal heterogeneity of climate warming. Therefore, this study will comprehensively and systematically explore the spatial and temporal heterogeneity of climate warming by analyzing intra-and inter-annual patterns of average temperature (AT) and their AV (amplitudes of variation) on different scales (10-day, seasonal, and annual), inter-annual changing trends, and the spatial concentration of warming pole based on 10-day air temperature data during 1961–2020/2017 (>50 years) from the available meteorological stations in the CTMR. The next section of the paper provides more details on data and methods. Section 3 is for results. The discussion then follows. The paper ends with conclusions. This study will improve the knowledge of regional climate change with global warming.

## 2. Data and Methods

### 2.1. Data

#### 2.1.1. Observed Data

The 10-day mean air temperature covering 1961–2010 and daily mean air temperature covering 2011–2020/2017 from 35 meteorological stations were analyzed in this study in the CTMR (Figure 1), which is provided by China Meteorological Data Service Centre (available online: http://data.cma.cn (assessed on 15 May 2021)). The quality of the data was firmly controlled before its issue, and homogeneity tests were also performed [26,35]. Quality control was then performed on 10-day and daily mean air temperatures, including visual inspection of single site or grid records and identifying outliers. Outliers were removed or corrected after identification. Only a small portion of the data required correction. Missing 10-day or daily data represented less than 2% of the total data. The data for a given day were estimated by extrapolating the average of the data from one or two preceding and following records. This study ignored any time series with missing data beyond one year. Then the 10-day mean air temperatures spanning 2011–2020/2017 were summarized by the daily air temperature during the corresponding period. Finally, the 10-day air temperature during 1961–2020/2017 from 26 meteorological stations in the CTMR were used, while data from 9 meteorological stations were omitted. Among these 26 meteorological stations, the data from 13 meteorological stations could span 1961–2020 and that from another

13 meteorological stations could only cover 1961–2017. The details of 26 meteorological stations are provided in Table 1.

**Table 1.** Descriptive information of the meteorological stations selected in the CTMR.

| Station Name | Station Code | Lon (°E) | Lat (°N) | Ele (m) | AAT_1961–2020/2017(°C) | AAT_1960s (°C) | AAT_2010s (°C) | AAV (°C) |
|---|---|---|---|---|---|---|---|---|
| Turpan | TP | 89.20 | 42.93 | 34.97 | 14.94 | 14.08 | 16.45 | **2.37 ↑\*** |
| Jinghe | JH | 82.90 | 44.60 | 320.87 | 8.07 | 7.45 | 8.79 | **1.35 ↑\*** |
| Caijiahu | CJH | 87.53 | 44.20 | 440.97 | 6.32 | 5.82 | 7.20 | **1.38 ↑\*** |
| Usu | US | 84.67 | 44.43 | 478.97 | 8.23 | 7.47 | 8.83 | **1.35 ↑\*** |
| Yining | YN | 81.33 | 43.95 | 663.20 | 9.29 | 8.58 | 10.20 | **1.63 ↑\*** |
| Shisanjianfang | SHSJF | 91.73 | 43.22 | 722.90 | 10.41 | 9.07 | 12.71 | **3.64 ↑\*** |
| Hami | HM | 93.52 | 42.82 | 738.20 | 10.20 | 9.86 | 10.88 | 1.02 ↑\* |
| Qitai | QT | 89.57 | 44.02 | 794.10 | 5.33 | 4.89 | 6.03 | **1.14 ↑\*** |
| Kumux | KM | 88.22 | 42.23 | 923.47 | 9.62 | 9.16 | 10.09 | 0.93 ↑\* |
| Korla | KL | 86.13 | 41.25 | 932.43 | 11.84 | 11.29 | 12.16 | 0.87 ↑\* |
| Urumqi | URMQ | 87.65 | 43.78 | 935.67 | 7.33 | 7.26 | 8.03 | 0.77 ↑\* |
| Kuqa | KQ | 82.97 | 41.73 | 1082.93 | 11.28 | 11.52 | 10.95 | −0.57↓\* |
| Yanqi | YQ | 86.57 | 42.08 | 1056.60 | 8.67 | 7.95 | 9.27 | **1.32 ↑\*** |
| Aksu | AKS | 80.23 | 41.17 | 1104.73 | 10.67 | 9.78 | 11.80 | **2.03 ↑\*** |
| Dabancheng | DBCH | 88.32 | 43.35 | 1104.77 | 6.74 | 6.26 | 7.09 | 0.84 ↑\* |
| Kalpin | KP | 79.05 | 40.50 | 1162.63 | 11.61 | 11.43 | 11.40 | −0.03 |
| Baicheng | BCH | 81.90 | 41.78 | 1230.00 | 8.04 | 7.60 | 8.76 | **1.16 ↑\*** |
| Kashi | KSH | 75.99 | 39.27 | 1291.20 | 12.18 | 11.67 | 12.68 | 1.01 ↑\* |
| Wenquan | WEQ | 81.02 | 44.97 | 1358.60 | 4.05 | 3.65 | 4.46 | 0.81 ↑\* |
| Barkol | BKL | 93.05 | 43.60 | 1675.37 | 2.34 | 1.20 | 3.65 | **2.45 ↑\*** |
| Yiwu | YW | 94.70 | 43.27 | 1728.60 | 4.06 | 3.46 | 4.71 | **1.25 ↑\*** |
| Balguntay | BLGT | 86.30 | 42.73 | 1739.47 | 6.60 | 6.13 | 6.27 | 0.15 ↑\* |
| Zhaosu | ZHS | 81.13 | 43.15 | 1853.40 | 3.51 | 2.89 | 4.26 | **1.37 ↑\*** |
| Wuqia | WUQ | 75.25 | 39.72 | 2176.90 | 7.38 | 6.71 | 7.70 | 0.99 ↑\* |
| Bayanbulak | BYBLK | 84.15 | 43.03 | 2459.27 | −4.29 | −4.67 | −4.20 | 0.47 |
| Tuergate | TEGT | 75.40 | 40.52 | 3506.40 | −3.19 | −3.82 | −2.61 | **1.21 ↑\*** |
| Average | | | | | 7.36 | 6.79 | 7.98 | **1.19 ↑\*** |

Lon: Longitude; Lat: latitude; Ele: Elevation; AAT: Annual average temperature; 1960s: 1961–1970; 2010s: 2010–2020/2017; AAV: Average amplitude of variation; \*: significant at 0.05 significance level; ↑ and ↓ means the upward trend and the downward trend, respectively; the bold means that the AAV value is higher than the 1.09 °C average of the global warming during 2011–2020 ($GWV_{2011–2020}$).

2.1.2. Geographical and Topographic Data

DEM (digital elevation model) data were derived from the Shuttle Radar Topography Mission website (available online: http://srtm.csi.cgiar.org (assessed on 18 August 2021)). The resolution of the DEM was 30 m. Geographical indices (longitude, latitude, elevation) and topographic indices (slope, aspect, surface roughness, surface northness, topographic relief, and rate of slope) were extracted station-by-station from the DEM of the CTMR on a grid scale based on the ArcGIS software. Here the surface roughness is generally defined as the ratio of surface area of a surface unit to its projection area on the horizontal plane in a specific area from the perspective of the geographic information system [34]. It is also a macroscopic index reflecting the surface morphology and one of the important indices affecting spatial heterogeneity. Northness, the product of the cosine of aspect and sine of slope, represents the influence of solar radiation. That is, the product of slope sine and slope direction cosine, which can be obtained by using a grid calculator. Topographic relief refers to the difference between the maximum elevation and the minimum elevation of all grids in a certain area. It can reflect that the undulation characteristics of the ground to some extent and is one of the indicators to describe the regional topographic characteristics. Topographic relief is an index reflecting the longitudinal changes of the terrain. The great topographic relief means the more obvious the variation degree of the surface in the vertical direction. Rate of slope, as the slope of slope, represents the second derivative of the

elevation of the ground surface with respect to the change of the horizontal plane, and can well reflect the profile curvature information to a certain extent [35,36].

### 2.2. Methods

To detect the warming within the CTMR, AV was defined as follows:

$$AV_i = AT\_2010s_i - AT\_1960s_i \tag{1}$$

where i represents the number of 10-day periods (i = 1, 2, 3, ... 36). $AT\_2010s_i$ means the AT from the ith 10-day period during 2011–2017/2020 in a certain meteorological station. $AT\_1960s_i$ means the AT from the ith 10-day period during 1961–1970 in a certain meteorological station. And $AV_i$ is the amplitude of variation from the ith 10-day period during 1961–2017/2020 in a certain meteorological station. This paper used AV to describe climate warming within the CTMR.

And AAV (the average value of AV within a year), $AV_{spring}$ (the average value of AV in spring, the 7th–15th 10-day periods), $AV_{summer}$ (the average value of AV in summer, including the 16th–24th 10-day periods), $AV_{autumn}$ (the average value of AV in autumn, including the 25th–33rd 10-day periods), and $AV_{winter}$ (the average value of AV in winter, the 1st–6th and 34th–36th 10-day periods) in a certain meteorological station could be calculated as followed:

$$AAV = \left( \sum_{i=1}^{36} AV_i \right) / 36 \tag{2}$$

$$AV_{spring} = \left( \sum_{i=7}^{15} AV_i \right) / 9 \tag{3}$$

$$AV_{summer} = \left( \sum_{i=16}^{24} AV_i \right) / 9 \tag{4}$$

$$AV_{autumn} = \left( \sum_{i=25}^{33} AV_i \right) / 9 \tag{5}$$

$$AV_{winter} = \left( \sum_{i=1}^{6} AV_i + \sum_{j=34}^{36} AV_j \right) / 9 \tag{6}$$

Without requiring normality or linearity, the non-parametric Mann-Kendall (MK) method was recommended by the World Meteorological Organization (WMO) to detect the long-term trends in a time series [37–39]. This method was widely used in the detection of trends of time series from hydrology and climatology [36,40–42]. Specific information about this method can refer to the research [23]. This paper used the MK trend test to check the changing trend of AV time series at annual and seasonal scales within the CTMR.

## 3. Results

### 3.1. General Characteristic of Warming

Table 1 showed values of 5 variables (AAT_1961–2020/2017, AAT_1960s, AAT_2010s, and AAV) from 26 meteorological stations in the CTMR. The AAT_1961–2020/2017 (AAT for short below) varied from −4.26 °C to 14.94 °C with the range of 19 °C in space. The high values of AAT were mainly located in the margin of the CTMR with lower altitudes, including TP, KSH, KL, KP, KQ, AKS, SHSJF, and HM, where AATs were over 10 °C and altitudes were below 1300 m asl. The low values of AAT were below 0 °C and appeared in regions with high altitudes over 2400 m asl, including TEGT (−3.19 °C) and BYBLK (−4.26 °C). According to the MK trend test, significant warming occurred at 23 meteorological stations and significant cooling at 1 meteorological station. There is no significant trend in KP and BYBLK (Table 1). Here, AAV means the difference between AAT_2010s and AAT_1960s and spanned from −0.57 °C to 3.64 °C in space. And the

average value of AAV was 1.19 °C, which was higher than GWV$_{2011-2020}$, 1.09 °C. The peak value of AAV occurred in SHSJF and the lowest value was obtained in KQ. AAVs over 1.09 °C were concentrated in 15 meteorological stations, 6 of which meteorological stations, including SHSJF (3.64 °C, over 3 times of GWV$_{2011-2020}$), Turpan (2.37 °C, almost 2 times of GWV$_{2011-2020}$), AKS (2.03 °C, 1.86 times of GWV$_{2011-2020}$), and so on, were located in regions with altitude below 1000 m asl, and another 6 meteorological stations, such as BKL (2.45 °C, over 2 times of GWV$_{2011-2020}$) were located over 1000 m asl. Among those 15 meteorological stations, climate warming exceeded the warming target (WT), 1.5 °C, at 5 meteorological stations, accounting for about 20% of the region. Negative values of AAV occurred in KQ and KP, where cooling existed.

To sum up, it was warmer in the margin of the CTMR with a lower altitude below 1300 m asl and colder in regions with a high altitude over 2400 m asl. Most of the CTMR became warmer, but the warming did not spread everywhere across the CTMR. The exceptions were in KQ (significant cooling) and KP (slight cooling). Amplitudes of variation were from −0.57 °C to 3.64 °C in space. Amplitudes of warming in about 46% of stations across the CTMR were higher than GWV$_{2011-2020}$, especially in SHSJF, TP, and AKS below 1000 m asl, and BKL over 1000 m asl.

### 3.2. Assessment of Geographical and Topographic Dependence

In order to assess the impact of topography on air temperature, the Pearson correlation between geographical and topographic variables, longitude (Lon), latitude (Lat), elevation (Ele), slope (Slo), aspect (Asp), surface roughness (Sur), northness (Nor), topographic relief (Tor), rate of slope (Ras), and AAT, AT$_{spring}$, AT$_{summer}$, AT$_{autumn}$, AT$_{winter}$, AAV, AV$_{spring}$, AV$_{summer}$, AV$_{autumn}$, and AV$_{winter}$ were conducted. It can be seen that elevation is negatively correlated with AAT, AT$_{spring}$, AT$_{summer}$, and AT$_{autumn}$ at 0.01 significance level (two-tailed test) and AV$_{winter}$ at 0.05 significance level (two-tailed test). There was significant negative correlation between AT$_{winter}$ and latitude at 0.01 significance level (two-tailed test) (Table 2). The impacts of other geographical and topographic variables on AT or AV were not significant. It means that the annual, springtime, summertime, and autumntime AT were elevation-dependent (Table 2 and Figure 2a–d). The significant regression equation between elevation and AAT, ATspring, ATsummer, ATautumn, ATwinter at 0.01 significance level (two-tailed test) was also showed in Figure 2. It could be seen that average air temperature in a year, spring, summer, and autumn would decrease by 0.5 °C, 0.5 °C, 0.7 °C, and 0.4 °C when the elevation increases by 100 m, respectively. The wintertime AT was latitude-dependent with a lapse rate of −1.56 °C per degree (Table 2 and Figure 2e).

**Table 2.** The Pearson correlation between geographical and topographic variables and AAT, AT$_{spring}$, AT$_{summer}$, AT$_{autumn}$, AT$_{winter}$, AAV, AV$_{spring}$, AV$_{summer}$, AV$_{autumn}$, and AV$_{winter}$ within the CTMR.

| | AAT | AT$_{spring}$ | AT$_{summer}$ | AT$_{autumn}$ | AT$_{winter}$ | AAV | AV$_{spring}$ | AV$_{summer}$ | AV$_{autumn}$ | AV$_{winter}$ |
|---|---|---|---|---|---|---|---|---|---|---|
| Lon | 0.07 | 0.08 | 0.29 | 0.02 | −0.19 | 0.32 | 0.31 | 0.30 | 0.20 | 0.10 |
| Lat | −0.20 | −0.20 | 0.06 | −0.20 | −0.51 ** | 0.23 | −0.02 | 0.27 | 0.15 | 0.16 |
| Ele | −0.75 ** | −0.79 ** | −0.91 ** | −0.77 ** | −0.32 | −0.24 | −0.23 | −0.02 | −0.09 | −0.39 * |
| Slo | −0.11 | −0.14 | −0.20 | −0.13 | 0.07 | −0.19 | −0.20 | −0.08 | −0.13 | −0.26 |
| Asp | 0.10 | 0.11 | 0.09 | 0.07 | 0.10 | −0.12 | −0.10 | −0.18 | −0.18 | 0.06 |
| Sur | −0.04 | −0.05 | −0.12 | −0.04 | 0.10 | −0.26 | −0.28 | −0.17 | −0.20 | −0.23 |
| Nor | −0.12 | −0.10 | 0.00 | −0.11 | −0.25 | 0.24 | 0.21 | 0.18 | 0.16 | 0.21 |
| Tor | −0.13 | −0.15 | −0.21 | −0.15 | 0.03 | −0.21 | −0.23 | −0.09 | −0.14 | −0.27 |
| Ras | −0.21 | −0.26 | −0.36 | −0.23 | 0.08 | −0.32 | −0.33 | −0.20 | −0.24 | −0.33 |

**: 0.01 significance level (two-tailed test); *: 0.05 significance level (two-tailed test).

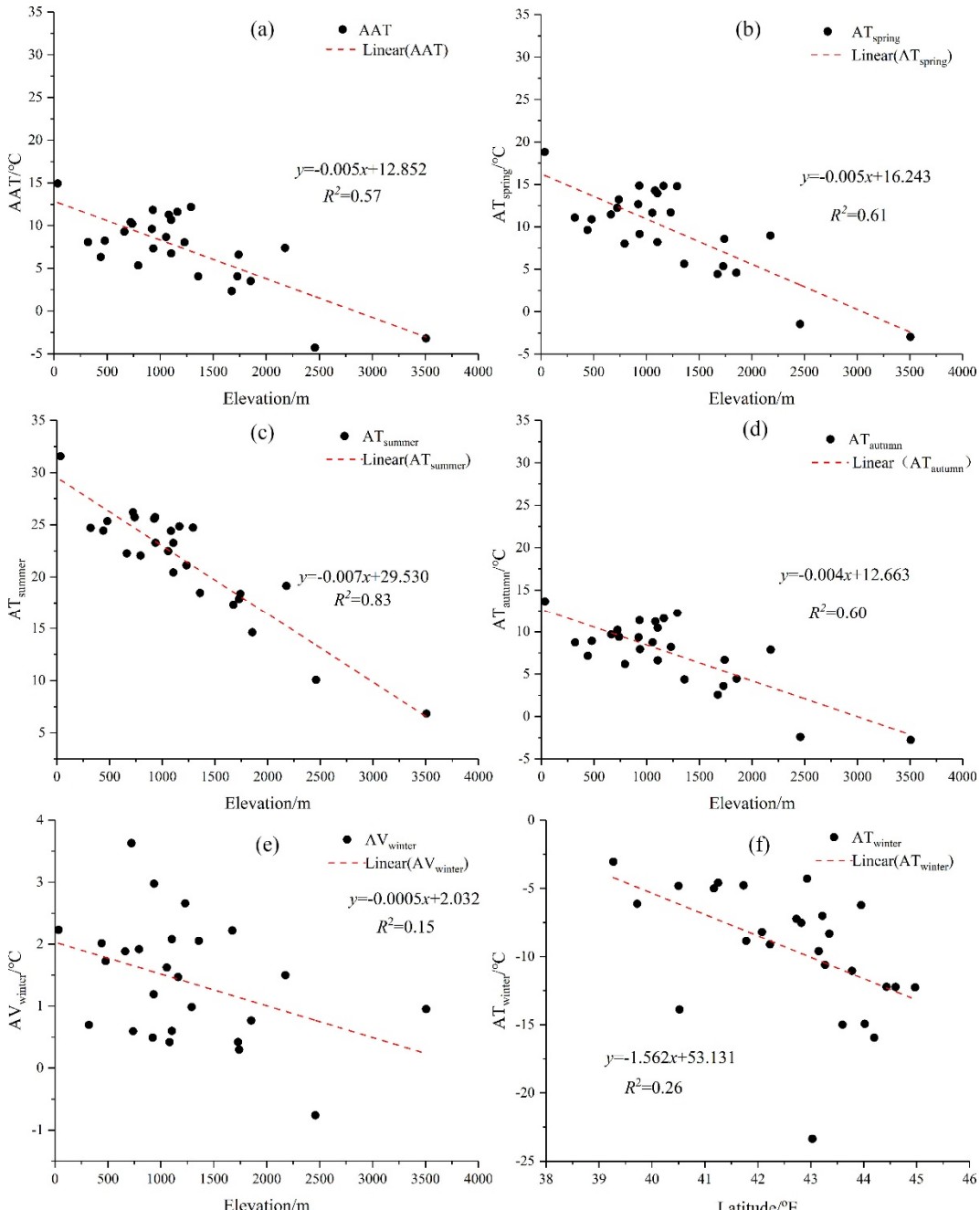

**Figure 2.** Changing pattern of AAT, AT$_{spring}$, AT$_{summer}$, AT$_{autumn}$, and AV$_{winter}$ at different elevations (**a–d**,**f**); and AT$_{winter}$ at different latitudes (**e**).

However, except for the significant correlation between elevation and AV$_{winter}$ with the negative correlation coefficient of −0.39 at 0.05 significance level, there were no significant correlations between elevation and AAV, AV$_{spring}$, AV$_{summer}$, and AV$_{autumn}$. That is to say, no significant elevation dependence existed in annual warming, springtime warming, summertime warming, and autumntime warming within the CTMR. In winter, the amplitude of variation decreased with the increase of elevation. The amplitude of variation of wintertime AT would decrease by 0.05 °C when the elevation increases by 100 m (Table 2 and Figure 2f).

In summary, annual, springtime, summertime and autumntime average air temperature and the amplitude of variation in winter were elevation-dependent with the lapse rate of −0.5 °C/100 m, −0.5 °C/100 m, −0.7 °C/100 m, −0.4 °C/100 m and −0.05 °C/100 m,

respectively. And average air temperature in winter was latitude dependent with the lapse rate of −1.56 °C per degree.

### 3.3. Temporal Heterogeneity of Warming

3.3.1. Intra-Annual Changing Pattern of Warming

Figure 3 showed the intra-annual changing patterns of the warming belt within a year (36 periods of 10 days); 1 for the average within the CTMR, 4 patterns for that from the stations at SHSJF, TLF, BKL, and AKS (those 4 meteorological stations were chosen as warming representatives for 26 meteorological stations for their high AV values). The average warming varied from 10-day period to 10-day period and peaked in the 10th 10-day period (the 4th 10−day period in spring), at 2.60 °C within the CTMR. The amplitudes of warming in the 19th 10−day period were higher than $GWV_{2011–2020}$. The negative values of AV occurred in the 8th, 15th, and 23rd 10-day periods (the 2nd 10-day period in spring, the 9th 10-day period in spring, and the 8th 10-day period in summer), which means that cooling happened (Figure 3a). Meanwhile, values of AV in the 11th 10-day period were already over the WT, which means the target was brought forward 10–12 years in those 10 days.

The warming belt fluctuated quite widely from 1.16 °C to 5.78 °C in SHSJF with an average belt width of 3.63 °C (Figure 3b). Additionally, the average amplitude of warming was higher than $GWV_{2011–2020}$ in each 10-day period. At the same time, except for the 15th 10-day period (the last 10 days in spring) within a year, amplitudes of warming in other 10-day periods were higher than the WT and peaked in the 10th 10-day period (the 4th 10-day period in spring) at 5.78 °C.

The situation of TLF was different and the warming belt fluctuated relatively smoothly from 1.13 °C to 4.77 °C with an average warming belt width of 2.81 °C. Similar to SHSJF, the average amplitude of warming was higher than $GWV_{2011–2020}$ in each 10-day period (Figure 3c). Except for the 1st 10-day period (the 4th 10-day period in winter) and the 15th 10-day period (the last 10 days in springs) within a year, amplitudes of warming in other 10-days were higher than the WT and peaked in the 6th 10-day period (the last 10 days in winter) with 4.77 °C.

For BKL, the warming belt fluctuated significantly with the range of 0.64–5.21 °C (Figure 3d); except for in the 27th, 30th, and 33rd 10-day period (all in autumn) within a year, amplitudes of warming in the other 33 10-day periods were higher than $GWV_{2011–2020}$. Values of AV were already over the WT from the 29th 10-day period among those 33 10-day periods. The maximum value of AV (5.21 °C) appeared in the 8th 10-day period (the 2nd 10-day period in spring).

In AKS, the warming belt also fluctuated largely from 0.63 °C to 3.89 °C (Figure 3e). Similar to BKL, except for the 13th, 16th, 18th, and 23rd 10-day periods (all in autumn) within a year, amplitudes of warming in the other 32 10-day periods were higher than $GWV_{2011–2020}$. Among those 32 10-day periods, values of AV were already over the WT from 25 10-day periods and peaked in the 10th 10-day period (the 4th 10-day period in winter) with 3.89 °C.

In summary, the warming changed from one 10-day period to the other and fluctuated diversely within the CTMR. The amplitudes of warming were higher than $GWV_{2011–2020}$ and the WT in over half of a year and in almost one-third of a year, respectively. SHSJF had the widest warming belt with the highest warming range of 1.16–5.78 °C, followed by TLF with the range of 1.13–4.77 °C, and then BKL with the range of 0.64–5.21 °C, and then AKS with 0.63–3.89 °C. The numbers of 10-day periods when the average amplitude of warming was higher than $GWV_{2011–2020}$ (1.09 °C) were 32, 33, 36, and 36 from AKS, BKL, TLF, and SHSJF, respectively. Numbers of 10-day periods when the average amplitude of warming was higher than the WT (1.5 °C) were 25, 29, 34, and 35 from AKS, BKL, TLF, and SHSJF, respectively.

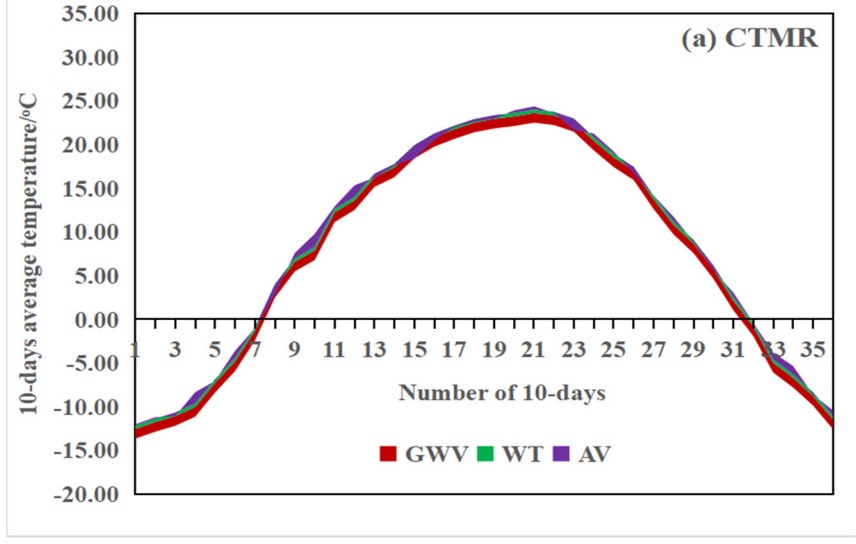

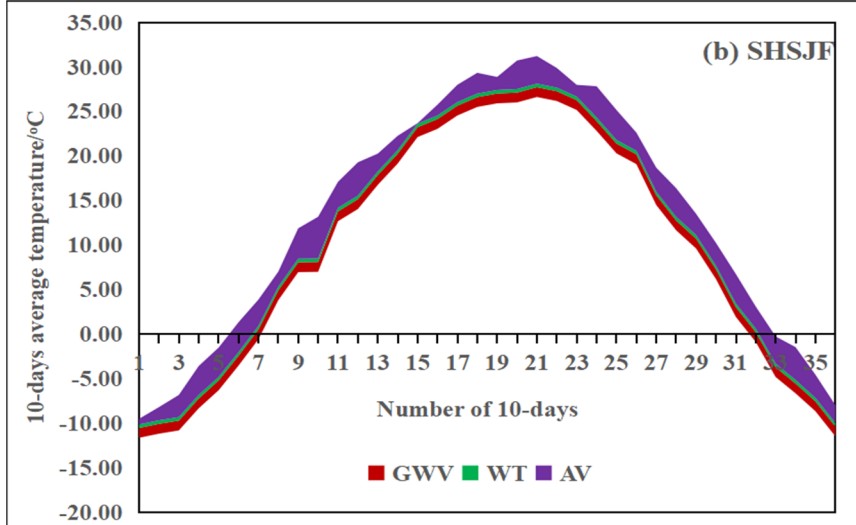

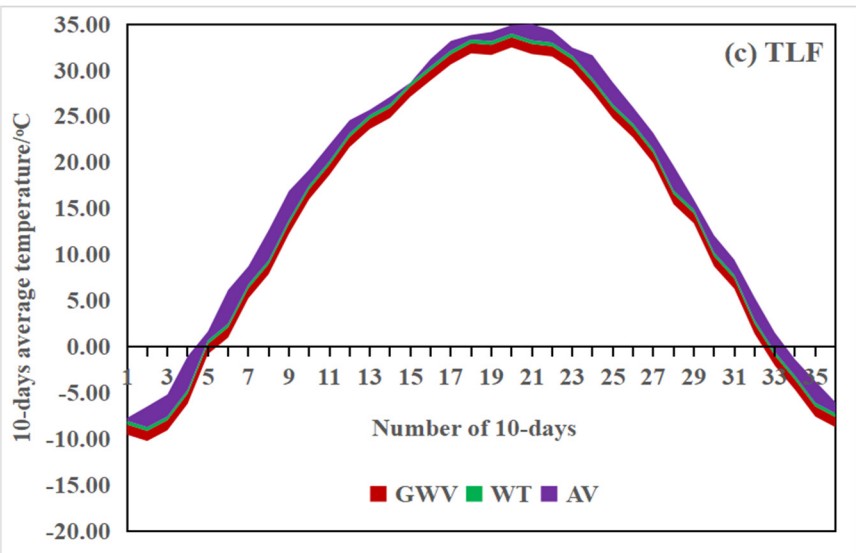

**Figure 3.** *Cont*.

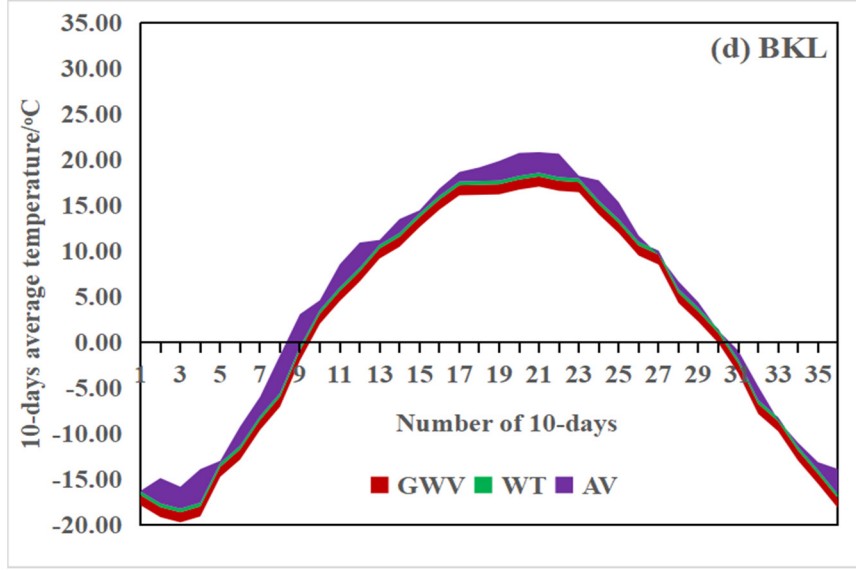

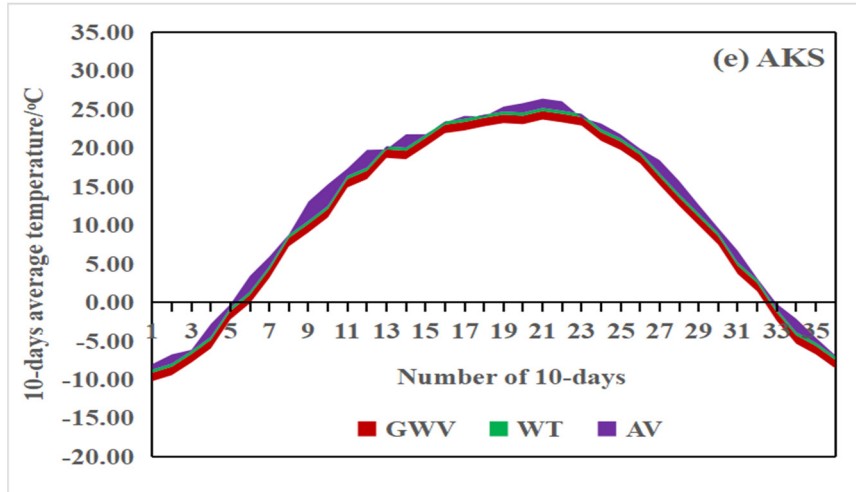

**Figure 3.** The intra-annual changing pattern of AV in $GWV_{2011-2020}$: the global warming during 2011–2020, 1.09 °C; WT: the warming target, 1.5 °C; AV: the amplitude of variation. (**a**) CTMR; (**b**) SHSJF; (**c**) TLF; (**d**) BKL; (**e**) AKS.

### 3.3.2. Classification of Warming

According to the MK trend test and values of AV in different seasons, overall warming (OW) is defined as warming that significantly occurred across the whole year. Warming in spring, summer, and autumn (WSSA) demonstrates that significant warming happened in spring, summer, and autumn. Cooling in summer and warming in winter (CSUWW) demonstrates that significant cooling and warming happened in summer and winter, respectively. The rest of the CTMR can be done in the same manner. In detail, the warming across the CTMR could be classified into nine types as follows: OW; WSSA; warming in autumn and winter (WAW); warming in summer, autumn and winter (WSUAW); warming in spring and summer (WSS); warming in summer and autumn (WSUA); warming in spring and autumn (WSPA); cooling in summer and autumn (CSUA); and cooling in summer and warming in winter (CSUWW). OW occurred at 14 stations, accounting for 54% of all stations in the CTMR. WSSA happened at 4 stations. WAW happened at 2 meteorological stations. WSUAW, WSS, WSUA, WSPA, CSUA, and CSUWW existed at 1 meteorological station each (Figure 4).

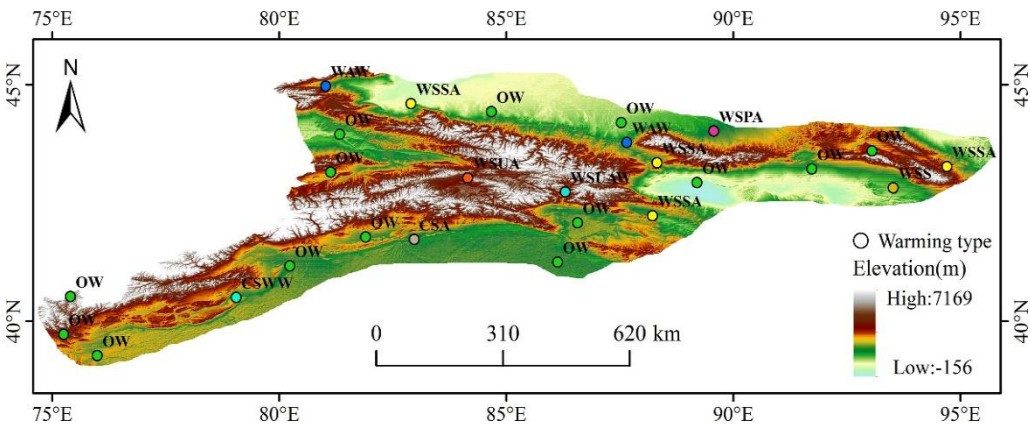

**Figure 4.** The spatial distribution of warming types in the CTMR. OW: overall warming; WSSA: warming in spring, summer, and autumn; WAW: warming in autumn and winter; WSUAW: warming in summer, autumn, and winter; WSS: warming in spring and summer; WSUA: warming in summer and autumn, WSPA: warming in spring and autumn; CSUA: cooling in summer and autumn; CSUWW: cooling in summer and warming in winter.

In summary, although warming spread year by year, there is temporal heterogeneity throughout the year. The amplitudes of warming in spring, autumn, and winter were higher than those in summer within the CTMR. Year-round warming was observed in 54% of the CTMR. A total 15% of the CTMR warmed in spring, summer, and autumn.

### 3.4. Spatial Heterogeneity of Warming

The spatial distribution of AV for annual, spring, summer, autumn, and winter across the CTMR is shown in Figure 5. The result was obtained by inverse distance weight interpolation. It can be seen that most of the CTMR warmed with different degrees throughout the year. The annual warming pole was located in the eastern part of the CTMR, especially in the Turpan-Hami basin, where AAV values reached above 3.64 °C. The Kuqa region, the Bayanbulak grassland, and the Kalpin hilly area were exceptions, where values of AAV were lower. Especially in the Kuqa region, the value of AAV was negative, which meaning that AAT was in a decreasing trend over the past 60 years.

The warming in spring was similar to the annual warming across the CTMR. The springtime warming pole was located in the Turpan-Hami basin with a peak value of 3.66 °C. The area with low values of AV in spring was larger than the AAV and it stretched from to Bayanbulak grassland to the north slope of the CTMR. However, the low values of $AV_{spring}$ were slightly higher than the AAV.

In summer, the warming pattern was also similar to the AAV across the CTMR. The difference occurred in the range of $AV_{summer}$ values. $AV_{summer}$ values varied from $-1.44$ °C to 3.35 °C, and both the peak and valley values of AV in summer were lower than those for the annual and the spring periods. Except for the Turpan-Hami basin, the Yili River valley was with high values of $AV_{summer}$, where the climate in summer was warmer than that of the annual and the spring periods.

The situation of AAV was similar in autumn across the CTMR. The values of $AV_{autumn}$ stretched to 3.89 °C and were higher than that for the annual, the spring, and the summer. The autumntime warming pole was located in the Turpan-Hami basin, with a peak $AV_{autumn}$ value of 3.89 °C. The cooling poles were also in the Kuqa region and the Kalpin hilly area with a negative $AV_{autumn}$ value of $-1.11$ °C.

The warming pattern in winter differed from the other seasons. Although the warming pole still existed in the Turpan-Hami basin, the cooling pole moved north to the centre of the CTMR. A new warming pole appeared in the Aksu region located in the southwest of the CTMR. $AV_{winter}$ values in winter ranged from $-0.75$ °C to 3.63 °C, which is consistent with the annual values.

In short, the spatial heterogeneity was clear for the warming pattern in different seasons across the CTMR. The warming amplitude in autumn was the largest and that in summer was the smallest. The warming area was larger than the cooling area. The warming pole in all seasons concentrated in the Turpan-Hami basin, and the cooling pole existed in the Kuqa region located in the south slope of the CTMR for most of the year.

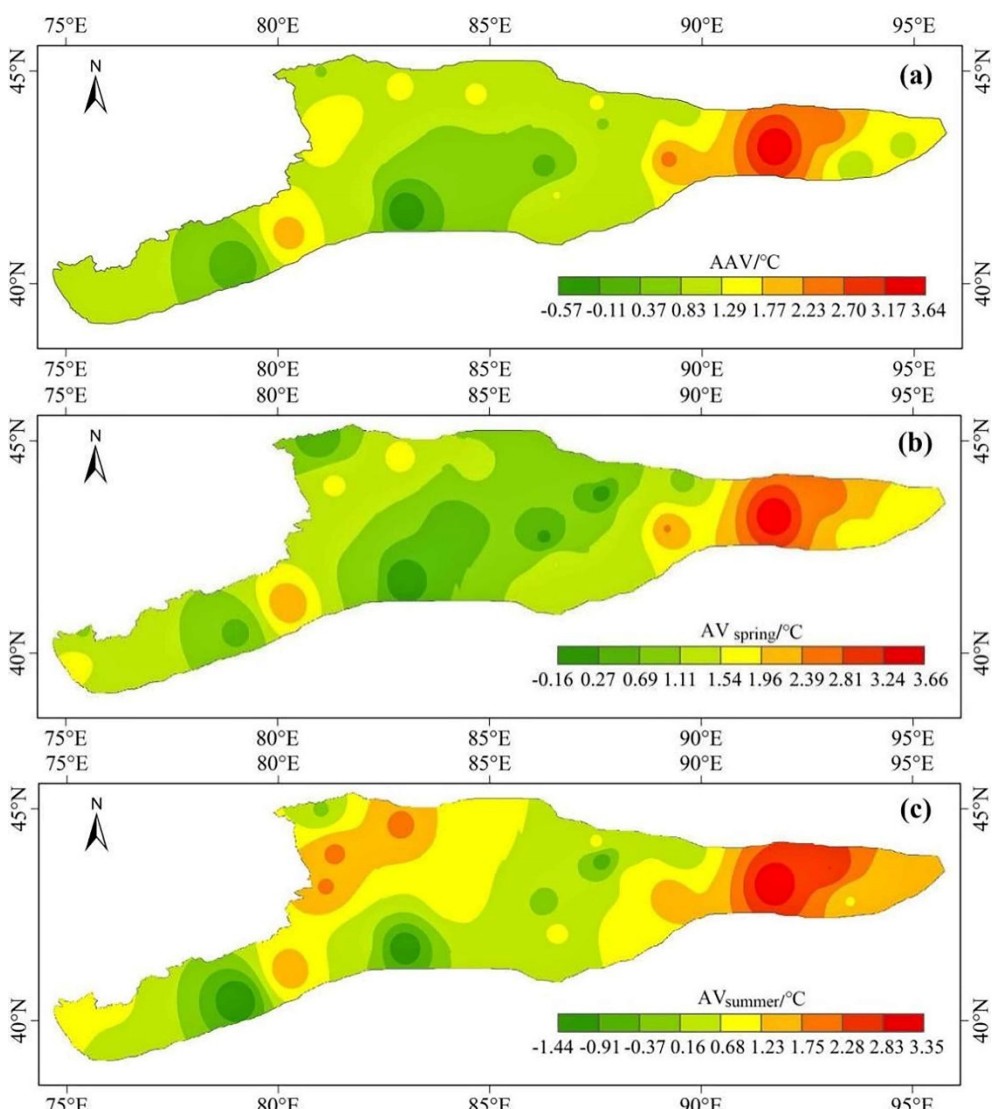

**Figure 5.** *Cont.*

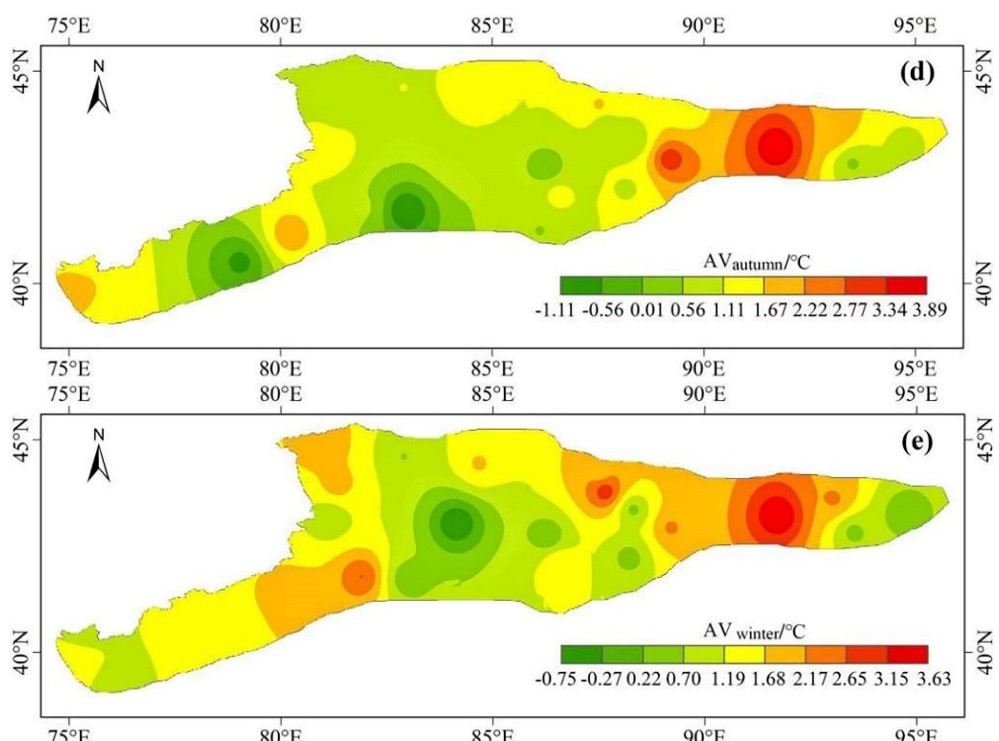

**Figure 5.** (**a**) Spatial distribution of AAV, (**b**) AV$_{spring}$, (**c**) AV$_{summer}$, (**d**) AV$_{autumn}$, and (**e**) AV$_{winter}$ within the CTMR.

## 4. Discussion

### 4.1. The Temporal Heterogeneity of Warming

Compared with the warming in spring, autumn, and winter, the amplitude of warming was small in summer across the CTMR. A total 54% of the CTMR showed warming in the whole year, and 15% of the CTMR was warming in spring, summer, and autumn. Seasonal cooling even happened in some places, BLGT, BYBLK, QT, KP, and KQ (Figure 6). In BYBLK, the winter air temperature varied from −27 °C to −17 °C and decreased by −0.76 °C during 2011–2017 relative to 1961–1970 without a significant downward trend at 0.05 significance level (Figure 6a). Although the air temperature in the summer at BLGT increased significantly at a 0.05 significance level during the whole period of 1961–2017, the values of summer air temperature decreased during 2011–2017 (Figure 6b). For QT, the air temperature in winter increased by 1.92 °C during 2011–2017 relative to 1961–1970. However, the trend was not significant at the 0.05 significance level (Figure 6c). In the summer of KP, the air temperature decreased with the rate of −0.26 °C/10a at 0.05 significance level and had decreased −1.17 °C during 2011–2017 relative to 1961–1970 (Figure 6d). In KQ, the air temperature in summer and autumn declined by −1.43 °C with the rate of −0.23 °C/10a and by −1.11 °C with the rate of −0.19 °C/10a at 0.05 significance level, respectively (Figure 6e,f).

The CTMR is located in the hinterland of Eurasia, with a vast territory and wide elevation difference. Its complex terrain has certain particularity for circulation. The subtropical high system is extremely developed over the Qinghai–Tibet Plateau in summer, forming the summer rain region in the CTMR [31]. The accumulated precipitation in summer had significantly increased by a rate of 4.4 mm/10a during 1961–2017 (Figure 7). The cooling effect of rainfall increased and perhaps offset the impact of global warming on the CTMR in summer. In addition, the CTMR is the source of numerous inland rivers, including the Yili River, Kuqa River, Weigan River, Akesu River, Hetian River, and Tarim River, et al. The increase of runoff in summer impacted by rising of atmospheric 0 °C level height could lower summer air temperature across the CTMR [43]. Perhaps these factors could explain why the amplitude of warming was smaller in summer than in other seasons.

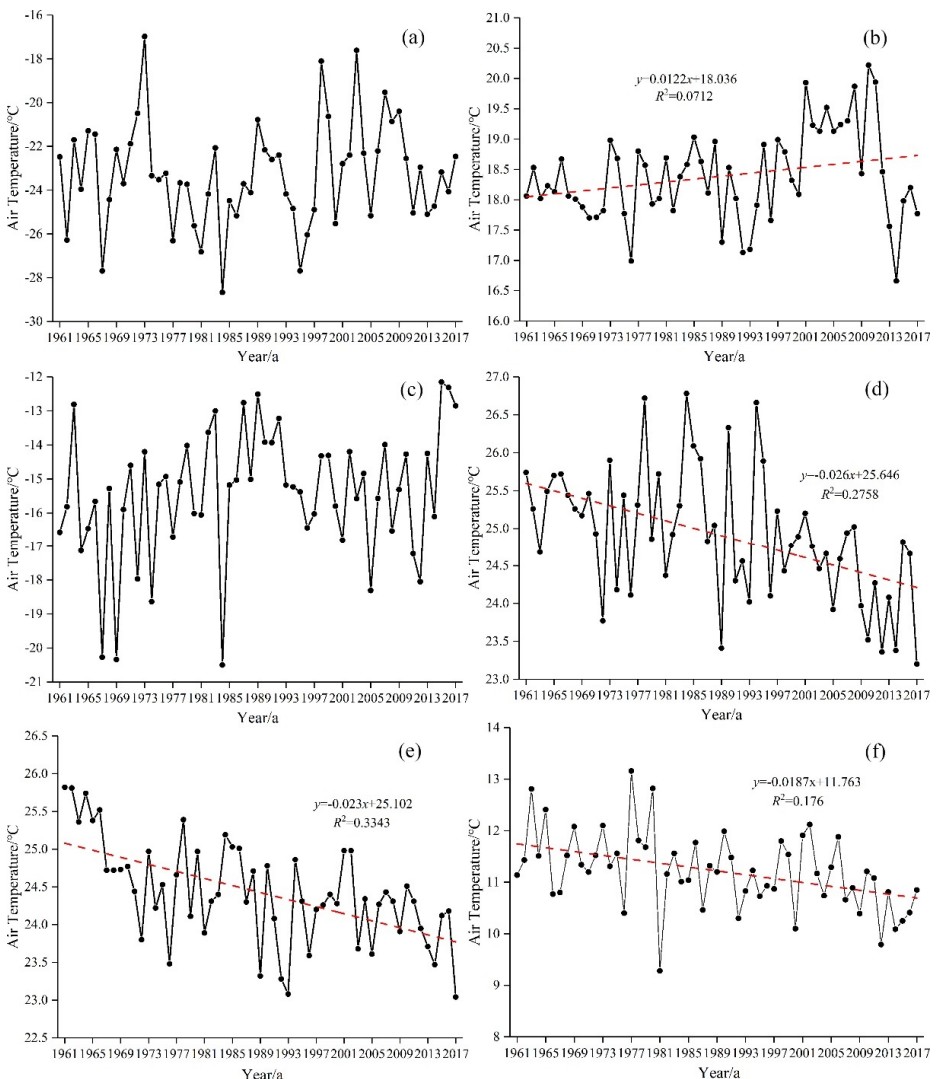

**Figure 6.** Exceptional time series of air temperature in the CTMR. (**a**) The wintertime changing trend in BYBLK; (**b**) the summertime changing trend in BLGT; (**c**) the wintertime changing trend in QT; (**d**) the summertime changing trend in KP; (**e**,**f**) the summertime and autumn changing trend in KQ, respectively.

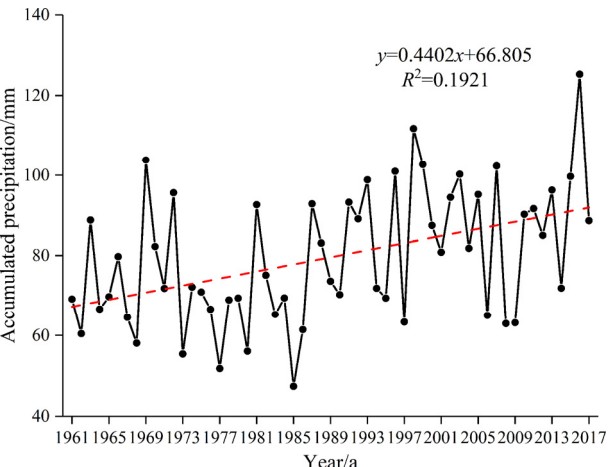

**Figure 7.** The time series of accumulated precipitation in summer during the 1961–2017 period across the CTMR (the precipitation data is from 31 meteorological stations within the CTMR).

*4.2. The Spatial Heterogeneity of Warming*

In more detail, the warming belt fluctuated diversely from one 10-day to the other across the CTMR. Within a year, the amplitude of warming was over $GWV_{2011-2020}$ in over half of the year and over the WT in almost one-third of a year, respectively. The warming belt in SHSJF was widest, with the largest warming range of 1.16–5.78 °C, which was beyond the whole belt of $GWV_{2011-2020}$ and most of the WT belt. That is to say, the warming in SHSJF was the extreme center across the CTMR.

There is a clear spatial heterogeneity of warming within the CTMR. Except of elevation and latitude, other topographic variables (longitude, slope, aspect, surface roughness, northness, topographic relief, and rate of slope) could not impact largely on air temperature or its AV. That is, the distribution of vertical and horizontal zonality existed in air temperature and its AV. In spring, the absolute value of lapse rate of average temperature was highest, over the global standard of surface air temperature lapse rate, −0.60–0.65 °C/100 m [44–46]. The wintertime AT was latitude dependent with the rate of −1.56 °C per degree. For AV, annual, springtime, summertime, and autumntime did not show significant vertical or horizontal zonality. Although using different data and different time scales, this point is similar to the research from Gao [47], at which the warming rate of mean air temperature in January and February (winter) showed prominent EDW features. In addition, such variability of lapse rate among different seasons may be influenced by the seasonal atmospheric features (e.g., air–moisture content, wind speed and direction, cloudiness, and radiation fluxes) [48–52].

Warming was more widespread than cooling across the CTMR. Significant cooling observed in KQ was caused by the significant decrease of the air temperature in summer and autumn with the rate of −0.233 °C/10a and −0.187 °C/10a at 0.05 significance level, respectively. Slight cooling in KP could result in a significant decrease of summer air temperature with the rate of −0.26 °C/10a at 0.05 significance level. In addition, the oasis irrigation in summer of KQ and KP may speed up the regional water cycle and bring more precipitation and low air temperature. The annual amplitude of warming in about 54% of the CTMR was higher than $GWV_{2011-2020}$, especially at the foot of the mountain with a low elevation below 1000 m asl. About 20% of the CTMR where the annual warming reached the WT, advanced at least 10 years compared to the expectation from IPCC [20].

The warm places were warmer across the CTMR. The warming pole in all seasons concentrated in the Turpan-Hami basin, the east part of the CTMR, where the air temperature itself was high. This may be attributed to the intense human influence at low elevations such as the Turpan-Hami basin due to easier reach, more human activities, and global warming. For example, the land surface temperature reached 80 °C in July 2021. The cooling poles existing in the center of the CTMR including the Kuqa region for summer and autumn.

This could be explained from perspectives of expansion of oasis, rising of atmospheric 0 °C level height and general atmospheric circulation in the Kuqa region. The area of oasis had expanded much faster during last 30 years [53]. It intensified the local water cycle and brought more local precipitation (the summer precipitation had increased significantly during 1961–2017 according to data from the meteorological station of Kuqa), and then local cooling in summer and even autumn occurred in Kuqa. In Kuqa, three rivers were distributed including the Kuqa River, Weigan River, and Tarim River. The rising of atmospheric 0 °C level height could cool down the local climate in Kuqa [43]. Besides this, the autumn cooling occurred in Eurasia and was likely influenced by the Pacific Decadal Oscillation (PDO) and Siberian high (SH) [32]. Kuqa, a part of Eurasia, was in the same situation as the rest of Eurasia in autumn.

The warming in the Yili River valley is a typical example of spatial and seasonal heterogeneity. The air temperature in summer increased faster than that in other seasons in the Yili River valley as we mentioned above. Global warming is the major reason for glacial recession in the Yili River basin [54]. If it continues, the rapid summer warming will necessarily accelerate the glacier ablation and retreat in the Yili River valley in the

future. Then it will lead to more problems or risks about local water resources and their management [55].

In addition, studies based on different length of time series from different data sources brought more uncertainty. Compared to the research based on ERA5-Land monthly averaged data from 1989–2018 and autumn air temperatures at weather stations from 2004–2018 across Eurasia [32], our study is based on the observed data of the recent 60 years and did not find a significant fall cooling trend. To reduce the uncertainty, a next step could focus on comparisons based on data from different sources with different temporal and spatial resolutions.

## 5. Conclusions

Global warming is unequivocal and will carry on for a long time in the future. The warming differs from one region to another with high spatial and temporal heterogeneity. The CTMR is quite a unique mountain system for its location and surroundings. It is extremely important for assessing climate change and ecological environment in northwest China, and even the whole country, due to its special geographical location and complex terrain. This paper analyzed in detail the spatial-temporal heterogeneity of warming across the CTMR based on the observed 10-day meteorological data of about 60 years. It was shown that annual, springtime, summertime, and autumntime average air temperature and the amplitude of variation in winter were elevation-dependent with the lapse rate of $-0.5$ °C/100 m, $-0.7$ °C/100 m, $-0.4$ °C/100 m, $-0.4$ °C/100 m, and $-0.05$ °C/100 m, respectively, which is likely affected by the seasonal atmospheric circulation (e.g., air-moisture content, cloudiness, wind speed and direction, and radiation fluxes). Most of the CTMR became warmer, but the warming did not spread every season and everywhere across the CTMR. On the temporal scale, the amplitudes of warming in more than half of year were amplified and higher than $GWV_{2011-2020}$. The temporal heterogeneity was reflected in that amplitudes of warming in spring, autumn, and winter were higher than that in summer. At a spatial scale, the warm places were warmer across the CTMR. It was warmer in lower altitudes below 1300 m asl such as the margin of the CTMR, where the warming was amplified by about 3.3 times compared to $GWV_{2011-2020}$. And it was colder in the Kuqa region. In a word, the large and complex spatio-temporal heterogeneity existed across the whole of the CTMR. This study greatly improved our understanding of spatio-temporal dynamics, patterns, and heterogeneity of warming across the CTMR and even northwest China.

**Author Contributions:** X.L. wrote, organized and revised the main text; B.Z. processed data, provided and revised figures, tables and references; Other group members (other authors) provided further insight, comments and editorial suggestions. All authors have read and agreed to the published version of the manuscript.

**Funding:** This work was financially supported by the National Natural Sciences Foundation of China (41761014, 42161025, 41871055 and 42101096), and the Foundation of A Hundred Youth Talents Training Program of Lanzhou Jiaotong University.

**Institutional Review Board Statement:** Not applicable.

**Informed Consent Statement:** Not applicable.

**Data Availability Statement:** Data are available upon reasonable request to the corresponding author.

**Acknowledgments:** We are grateful to the reviewers for their valuable comments on this manuscript, we also thank the editor for his contribution to the manuscript.

**Conflicts of Interest:** The authors declare no conflict of interest.

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
