# Peer review of "Spatio-Temporal Heterogeneity of Climate Warming in the Chinese Tianshan Mountainous Region"

_water, doi:10.3390/w14020199_

Round 1

Reviewer 1 Report

In lines 48 to 50, the sentence “Climate warming has more severe climate warming, while other areas have climate warming, and some places may even cool down” is incomprehensible, rewrite.

It could be something like: “Climate warm has more severe climate warming, while other areas have milder climate warming, and some places may even cool down.”

In lines 75 to 77, the sentence “And the warming also differed from year to year and season to season, with large temporal heterogeneity. both seasonal and annual warming was observed in Northwest China” is incomprehensible, rewrite.

Was it something like that? “Furthermore, warming was different from year to year and from season to season, with large temporal heterogeneity, with both seasonal and annual warming being observed in Northwestern China.”

Line 78 is missing a ‘period’ after [18]: “rate [18] With”.

Figure 1 presents a lot of information that makes visualization difficult. I suggest that some way be adopted to highlight the name of the stations, it can be putting in bold or changing the font color, or even putting the name of the station on a white background over the image. As it is, it's almost impossible to recognize them all.

In table 1 there are some values in bold and there is no explanation in the table of what this means. Either remove the bold or explain in the footnote, together with the other explanations.

In Figure 2, in the ATwinter versus Latitude graph (e), I suggest that the connection of the points is done in ascending order of Latitude. As it stands (I think it follows the order of altitude) the line has no clear meaning for the reader.

Figure 4 is already visually less 'polluted' than figure 1, but, as mentioned earlier, it is necessary to improve the identification of the abbreviations of the stations, which in some cases are not fully visible. Look for example WSAW and WSSA.

Reviewer 2 Report

This study presents an overview of the spatio-temporal heterogeneity of climate warming in the Chinese Tianshan mountainous region, based on observational data from 26 weather stations. Authors have identified substantial spatial heterogeneity of the warming in this region. In particular, they found that weather stations located in a warm part showed strong warming, while at the weather stations located at high altitudes, the mean temperature even somewhat dropped between 2010s and 1960s. This conclusion may be interesting from the point of view of glacier melting analysis. However, the authors do not provide any information on this topic. It is important since the main topic of this journal is water science and technology, including the ecology and management of water resources. Authors also do not compare this result with previously published ones for other high mountain systems in Eurasia.

In addition, there are several concerns mainly related to the description of the used methods and data visualization (figures and maps). Therefore, i recommend major revision of this manuscript.

MAJOR COMMENTS

  1. In general, the authors have found strong spatial heterogeneity in warming intensity between neighboring weather stations, but do not give an explanation for this difference. The discussion mainly lists what has already been given in the results section.
  2. Any information characterizing the degradation of mountain glaciers in CTMR region during the study period (1961-2018), based on previously published results (e.g. https://doi.org/10.1155/2015/847257, 11928/j.issn.1001-7410.2021.05.02) should be added to the discussion, to show how the spatial and seasonal heterogeneity in observed climate warming found by the authors correspond to the observed change in glaciers
  3. In the discussion section, it is needed to compare the main finding of this study with similar published results for other high mountain systems in Eurasia.
  4. If the authors also provide any information on the observed changes in precipitation amount in CTMR during 1960-2020, this could substantially improve the manuscript.

Other specific comments

Lines 48 – 50: rewrite sentence for clarity

Figure 1: it is better to not extract the DEM by the CTMR border, and add the CTMR border as a vector layer. Replace “station to “weather station” in the legend

Lines 146 – 148: longitude, latitude, elevation are not topographic indices

Line 148: which DEM was used? SRTM or other? It is needed to add this information (including spatial resolution of the DEM and data source) to the text

Line 152: Replace ‘indexes’ to ‘indices’ throughout the text

Subsection 2.2.2: Mann-Kendall trend test is well-known and widely used by climatological community. So, I'm not sure what to describe the calculation process, you can only put a link to a relevant source, and there is no need to separate subsection 2.2 (Methods) to 2.2.1 and 2.2.2

Subsection 3.1. This paragraph rather lists the values of AAT and AAV that are already given in table 1. Therefore, i recommend to strongly reduce it, leaving only key points

Line 228: Pearson correlation, or other coefficient was used?

Lines 239 – 252: Here it is possible not to give the equations еthemselves, since only trend coefficients are of interest

Line 246: Replace ‘0.05 significant level’ to ‘0.05 significance level’

Table 2: what correlation coefficient was used?

Figure 2: I strongly recommend remove these dark blue lines (connecting dots) from all sub-figures, and leave only linear trends. It is also better to move the axes intersection on the graphs so that all points are to the right of the axis ’Y’.

Figure 4: the same comments as for Figure 1

Line 343: replace ‘was shown’ to ‘is shown’

Figure 5: what interpolation method was used?

Line 384-392: This paragraph mainly duplicates the information from Introduction section

Figure 6: Labels on figures such as BYBLK_winter should be removed, since they duplicates the information from figure caption

Line 435: replace ‘popular’ to more appropriate word

Line 451: 80ËšC is too high for air temperature. Maybe this is soil surface (skin) temperature?

Line 453 – 455: I recommend delete these two sentences about the arctic amplification, and replace them to the information on other mountainous regions in Eurasia (see Major comments)

Round 2

Reviewer 2 Report

The authors took into account most of comments and improved the manuscript. However, there are still a few comments that need to be corrected before the manuscript can be accepted for publication.

  1. Delete “Legend” word from Figure 1 and Figure 4.
  2. Line 149: DEM abbreviation should be introduced for the first mention (Digital Elevation Model)
  3. Line 219: “pearson” should be “Pearson” (with a capital letter) here and throughout the text
  4. Line 231: replace “significant level” to “significance level” here and throughout the text
  5. Line 334 – replace “Kring” to “Kriging”. It is also of note that Kriging interpolation (in ArcGIS Geostatistical Analyst) has many settings such as kriging method itself (Ordinary kriging, Simple Kriging, Universal Kriging), semi-variogram model selection, number of lags, data normalization method, trend removing, and many other. All these settings must be justified as they depend on the data properties and strongly affected the result of interpolation. Therefore, if authors performed interpolation with ArcGIS Spatial Analyst, I recommend use a simpler method, for example local polynomial interpolation or inverse-weighted distance method with an exponent close to one
  6. Line 394 – 396: warming in summer is slower than in other seasons in a significant part of Eurasia. Therefore I am not sure if the authors' assumption that the warming will take place more slowly due to summer rainfalls is correct
  7. Line 396: replace “the the” to “the”
  8. The results of interpolation (fig. 5) may be also compared with the data of WorldClim 2.0 project, more precisely with Historical Monthly Weather Data (https://www.worldclim.org/data/monthlywth.html), since they also contains average annual temperature for decades 1960-1969 and 2010-2018 in Geotiff format with high spatial resolution. Moreover, these data may be even more correct comparing with presented results, since elevation-dependent regression was used for interpolation. This comparison may be added to Results or Discussion section.
